# The Mental Health Impact of the COVID-19 Pandemic Second Wave on Shielders and Their Family Members

**DOI:** 10.3390/ijerph19127333

**Published:** 2022-06-15

**Authors:** Jo Daniels, Hannah Rettie

**Affiliations:** 1Department of Psychology, University of Bath, Bath BA2 7AY, UK; hannah.rettie@nbt.nhs.uk; 2North Bristol NHS Trust, Bristol BS10 5NB, UK

**Keywords:** anxiety, coronavirus, contamination, behaviour, shielding

## Abstract

In March 2020, individuals shielding from coronavirus reported high rates of distress. This study investigated whether fear of contamination (FoC) and use of government-recommended behaviours (GRB; e.g., handwashing and wearing masks) were associated with psychological distress during February 2021. An online cross-sectional questionnaire assessed psychological distress in three groups (shielding self, shielding other/s, and control), and those shielding others also completed an adapted measure of health anxiety (α = 0.94). The sample (*N* = 723) was predominantly female (84%) with a mean age of 41.72 (*SD* = 15.15). Those shielding (self) demonstrated significantly higher rates of health anxiety and FoC in comparison to other groups (*p* < 0.001). The use of GRB was significantly lower in controls (*p* < 0.001), with no significant difference between the two shielding groups (*p* = 0.753). Rates of anxiety were higher when compared to March 2020 findings, except for controls. Hierarchical regressions indicated FoC and GRB accounted for 24% of variance in generalised anxiety (*p* < 0.001) and 28% in health anxiety, however, the latter was a non-significant predictor in final models. Those shielding themselves and others during the pandemic have experienced sustained levels of distress; special consideration must be given to those indirectly affected. Psychological interventions should account for realistic FoC and the impact of government-recommended health behaviours, as these factors are associated with distress in vulnerable groups and may extend beyond the pandemic. Future research should focus on longitudinal designs to monitor and better understand the clinical needs of those shielding, and those shielding others post-pandemic.

## 1. Introduction

Coronavirus infectious disease (COVID-19) was first detected in Wuhan, China, in December 2019. Since then, the respiratory coronavirus has spread worldwide, declared a ‘public health emergency of international concern’ in January 2020. The virus is spread through an infected person coughing or sneezing and will cause mild to moderate symptoms in most individuals infected. Symptoms include fever, tiredness, and a persistent dry cough [1]. In July 2020, Remdesivir was authorised as the first COVID-19 treatment in the EU [2] and in the UK [3,4], allowing for vaccination rollout to begin on the 8th of December, ahead of most other countries [5].

During the period of vaccine and treatment development, those in clinically vulnerable groups were adversely affected. Shortly after the declaration of the pandemic status of COVID-19, the UK government identified those who were ‘clinically vulnerable’ and advised them to shield themselves. These individuals were encouraged to work from home, avoid contact with others, and ask others to do their shopping when possible [6]. From 1 August 2021, government shielding guidance ended; however, due to the further waves of COVID-19 and despite the prioritised rollout of vaccinations to those recognised as clinically vulnerable, many individuals still chose to continue shielding and take additional precautions.

The mental health impact of COVID-19 is now well documented, with findings from numerous systematic reviews showing lower levels of psychological well-being and higher rates of anxiety and depression during the initial phase of the first wave (March–April 2020) in comparison to pre-pandemic rates [7,8]. However, the impact appears to have been temporary for most, with a review of longitudinal cohort studies finding that by May–July 2020, rates of distress in the general population had returned to near pre-pandemic levels [8].

Despite an abundance of research examining the mental health impact of COVID-19 within the general population, research involving those advised to shield is still emerging. Previous research by Rettie and Daniels [9] found that individuals who were categorised in government-defined ‘vulnerable groups’ were significantly more likely to display clinical levels of generalised anxiety and health anxiety in comparison to the rest of the general population during early UK lockdown. Similarly, Robinson et al.’s review [8] noted a larger increase in distress in those with pre-existing physical health conditions during the first wave of COVID-19 in comparison to the general population.

This may be associated with the degrees of uncertainty faced by this group; the model of uncertainty distress [10] suggests that increased perceived and actual threat during COVID-19 is likely to result in greater psychological distress. This was highlighted in Philip et al.’s [11] qualitative study with people who have long-term respiratory conditions, where participants reported increased vulnerability to COVID-19 and concerns about the likelihood of survival if they were to contract it. These feelings of elevated anxiety for one’s physical health have been replicated across other studies [12,13], and research exploring the experiences of women with breast cancer reported the negative impact delays to treatment and reduced access to services had on participants’ emotional well-being [13,14]. Many patients reported that disruptions to medical services resulted in being abruptly discharged, treatment programmes being put on hold, and professional support being retrieved [15]. These disturbances frequently led to cancelled appointments, lack of communication, delays in responses to requested help, and inconsistencies in delivered information during lockdown [16]. Although remote support had relative success, patients reported feeling emotionally distressed and abandoned by medical services [14,17]. Emotional distress was also found to be associated with reduced treatment compliance, which could engender disease progression and mortality [18].

A rapid review of the psychological impact of quarantine reported various negative effects (e.g., anger, stigma, and post-traumatic stress symptoms) that could be long-lasting [19]. These negative psychological outcomes may have been exacerbated by the lack of routine and normality, resulting in a loss of purpose and restlessness in those shielding [15]. Reduced social interactions and loneliness were also interpreted as significant factors of depressive symptoms and anxiety levels reported in shielders [20]. Additionally, damage caused to shielders’ self-identity from being labelled ‘vulnerable’ and being ‘othered’ by the media heightened fears of social stigma in being blamed for lockdown restrictions [16,21].

Exploring predictors of distress has been identified as a key research priority during COVID-19 [20], with predictors relevant across the general population fairly well researched. While unexplored in this group, fear of contamination is likely to be a key factor in relation to psychological distress during COVID-19 for shielders. Previous research during the H1N1 pandemic found that perceived severity and likelihood of contamination predicted H1N1-specific levels of anxiety in students [22], and contamination fear has been found to predict safety-seeking behaviour during COVID-19 [23]. For individuals who are shielding and more vulnerable to the effects of COVID-19, fear of contamination may be amplified with over-engagement in safety-related behaviour in an attempt to manage this (e.g., excessive hand washing and avoiding others), which, according to the cognitive behavioural model of anxiety, would result in increased psychological distress. Distress may have been more elevated in individuals belonging to ethnic minority groups or from low-income households due to higher perceived risks of mortality associated with these groups [14,16].

In taking a wider perspective, the impact of COVID-19 on family members living with shielders should also be considered. Many family members and those closest have chosen to shield a vulnerable person or take additional precautions, yet this population has been largely neglected in both policy and research. It is unclear what the mental health impact on this group might be and whether they are engaging in similar precautionary measures. Shielding family members involved frustration associated with the lack of personal space, and extreme loneliness and discouragement without the help of family and friends was expressed [15,17], yet there is little to explore this. Fancourt et al. [24] found that during early lockdown, 30% of individuals across the UK general population were worried about family or friends who lived inside their household, with research also reporting that witnessing clinically vulnerable individuals’ decline in physical and mental health also increased anxiety and depressive symptoms in family members [17]. Investigating levels of distress and anxiety in family members will help uncover whether family members of those shielding are also a group vulnerable to increased mental health distress and will need additional support during this time. In the words of a participant in our previous study [9], ‘I am anxious because I am shielding someone else, not for myself.’ This level of ‘vicarious’ health related anxiety warrants exploration.

It is evident that there continues to be a gap in the literature pertaining to the psychological health and needs of those shielding themselves or others, with many studies to date being small scale, qualitative, and limited to the person who is clinically vulnerable. While emerging studies have explored the nature of the distress, further work is needed to identify potential targets for treatment and the commonality of experience, particularly for this potentially new complex group who shields others, in order to gain a better understanding of the impact of what one might term as ‘vicarious health anxiety’.

Currently, Cognitive Behavioural Therapy (CBT) is recommended for common mental health problems [25]; however, despite previous research exploring cognitive and behavioural factors in previous pandemics [26,27,28,29], the utility and adaptation of the CBT model remains largely unexplored when considering the needs of those shielding. Given the established distress in this group and the move towards accepting a state of endemic COVID-19, it is vital we understand the nature of distress in those shielding, how this differs between those shielding self or others, and how we could better tailor current evidence-based interventions.

### Study Aims

The aims of this study were to investigate the mental health impact of the COVID-19 pandemic second wave on shielders and their family members, and to examine key factors that may be relevant to the adaptation and development of mental health interventions for these vulnerable groups and those that support them. The primary research questions were:What is the incidence of generalised and health anxiety in those shielding themselves and others during the second wave of COVID-19?To what degree do contamination fears and COVID-19-driven health behaviours influence psychological distress?How does this differ when comparing those shielding self, those shielding others, and those who have never shielded?

## 2. Materials and Methods

### 2.1. Ethical Approval

Ethical approval for the project was obtained from the University of Bath ethics committee (PREC Reference Number 20-260).

### 2.2. Design

This study consisted of a cross-sectional online questionnaire study over a four-week period in February 2021 in the second wave of the COVID-19 pandemic, when the UK was in a national ‘lockdown’.

### 2.3. Participants and Procedure

Anyone above age 18 who lived in the UK and was either shielding themselves or shielding others (only) was invited to participate in the study. A control group of individuals who had never shielded was also collected. Individuals taking part could have shielded/be shielding due to concerns about their own physical health or to protect the health of someone they live with. Snowball sampling was used to maximize recruitment during the limited time period through social media (e.g., Facebook and Twitter) and email distribution lists. Individuals chose to participate by self-accessing the study link and providing consent. All participants were provided with study and debrief information following questionnaire completion.

Of the 852 participants in the survey, 723 of these completed the main study questionnaires fully (84.9%). A total of 390 participants were shielding themselves, 69 were shielding others, and 264 were not shielding currently. The incomplete surveys were excluded listwise from any further analyses as chi-square analyses, and one-way ANOVAs determined that, except for age, there were no significant differences between the demographics of these individuals and those who fully completed the main questionnaires. The mean age of excluded surveys (*M* = 47.09) was significantly older than the included surveys (*M* = 41.72; *F*(1, 766) = 6.78, *p* = 0.009).

The demographic information of the 723 participants can be found in Table 1. Participants were considerably spread out across the UK but somewhat concentrated around Southwest England, Southeast England, Northwest England, and Wales.

The mean age of participants was 41.72 years (*SD* = 15.15), with 13.8% of individuals living alone. A total of 86 participants (11.9%) believed that they currently or previously had COVID-19, with 429 (59.3%) of individuals reporting that a close friend or family member had been diagnosed with COVID-19. Included in the sample were 268 participants (37.1%) indicating they had received the COVID-19 vaccine. Two hundred and forty-two participants (33.5%) reported having a pre-existing mental health condition, which primarily consisted of anxiety (18.6%), depression (19.4%), or co-morbid anxiety and depression (19%).

Of the included participants, 481 individuals had shielded for an average of 11.56 (*SD* = 2.80) months. Details regarding the physical health conditions of those shielding themselves are shown in Table 2.

### 2.4. Measures

The 10-item contamination subscale of the Padua Inventory [30] was used to measure fear of contamination. Previous research using this subscale during COVID-19 found adequate internal consistency [23] (α = 0.82), and higher scores indicated greater contamination fear. In the current sample, internal consistency was excellent (α = 0.92).

The GAD-7 measure of anxiety has shown excellent internal (α = 0.92) and test–retest reliability (*r* = 0.83) in a clinical sample, alongside criterion and construct validity [31]. A clinical cut-off score of 10 and above was used [32], and in the current sample, internal consistency was excellent (α = 0.91).

The Short Health Anxiety Inventory (HAI) is a 14-item measure assessing health anxiety [33] based on the cognitive-behavioural model of health anxiety. The HAI had shown internal consistency (α = 0.86) and convergent and divergent validity in a non-clinical sample [34], and a clinical cut-off score of 18 or above was used. Internal consistency for the HAI in this sample was excellent (α = 0.90).

In the absence of a suitable measure, the HAI was adapted for use in the current study to measure ‘vicarious’ health anxiety. That is, anxiety focused on the health of another. The only changes made were to the ‘subject’ of the questions, i.e., where beliefs were elicited about ‘self’, now relate to the shielded ‘other’ (e.g., I do not worry about the health of my family member.). Internal consistency of the vicarious HAI (α = 0.94) in the current sample was excellent; tests of convergent validity indicated that the vHAI was related but distinct from health anxiety (*r* = 0.438) and generalised anxiety (*r* = 0.436). As this was a purpose-adapted measure (restricted to changing the ‘subject’ rather than content), a Principal Component Analysis (PCA) was performed to examine the lower-factor structure. Assumptions of the PCA were assessed, and all variables had a correlation coefficient above 0.3 on the correlation matrix. The overall Kaiser–Meyer–Olkin (KMO) measure of sampling adequacy for the scale was 0.923 [35] (classified as ‘marvellous’), and Bartlett’s test of sphericity was significant (*p* < 0.001), indicating that the questionnaire was factorisable. The scree plot, percentage variance, and eigenvalues were used to determine how many factors should be retained. The PCA indicated a potential two-factor solution, which explained 63.37% of the total variance, with most variance explained by the first factor (55.03% and 8.34%, respectively). However, visual inspection of the scree plot indicated that only one component should be retained, and a varimax orthogonal rotation of the two-factor solution indicated a complex structure with eight of the fourteen factors loading on both components (based on criteria of 0.3 as a salient loading). It was concluded that the data could not be factorised, and a one-factor structure was retained.

A seven-item measure was developed to measure COVID-19-related health behaviours for use in the current study. Items were devised by (a) consulting safety behaviour scales used in published COVID-19 research [23,36], (b) identifying recommended behaviours from government guidelines [37], and (c) general behavioural trends reported during the COVID-19 pandemic (e.g., stockpiling) [38]. Participants were asked to score on a scale of 1–5 whether they do these behaviours less than government guidance, in line with government guidance, or more than government guidance. Previous behaviour scales were not used as many of the behaviours described in these scales were recommended by the government at the time of lockdown (e.g., mask wearing), and the questionnaires did not capture whether they were used excessively. The internal consistency of this measure in the current study was ‘questionable’ (α = 0.61); however, as 85% of single-item correlations were significant, with each item significantly correlating with between four to six of the other items in the scale, the scale was still considered to capture a single construct [39].

A PCA was performed on the COVID-19-related health behaviour scale to examine the lower-factor structure. Assumptions of the PCA were assessed, and all variables except ‘getting COVID-19 tests’ and ‘attending clinical appointments’ had a correlation coefficient above 0.3 on the correlation matrix. The overall Kaiser–Meyer–Olkin (KMO) measure of sampling adequacy for the scale was 0.709 [35] (classified as ‘middling’), and Bartlett’s test of sphericity was significant (*p* < 0.001), indicating that the questionnaire was factorisable. The PCA indicated a three-factor solution, which explained 65.51% of the total variance, with most variance explained by the first factor (30.85%, 17.29%, and 13.37%, respectively). A varimax orthogonal rotation indicated that the variables exhibited a simple structure and were based on a criterion of 0.3 as a salient loading, all items loaded singularly onto one factor. Factor one was composed of three items with salient loadings and was labelled ‘government guidance related behaviours’ because all three factors related to the main government guidance of ‘hands, face, space.’ Factor two consisted of ‘stocking up on essentials’ and ‘checking the internet for information on COVID-19’ and was labelled ‘excessive/preparatory behaviours’ because this factor consisted of behaviours that were preparatory in nature and not recommended in government guidance, so it could be considered primarily safety behaviours used to manage feelings of anxiety and uncertainty. Finally, factor three contained two items and was labelled ‘medical related behaviour’ as it consisted of items relating to accessing healthcare.

### 2.5. Data Analysis

Data analysis was completed using SPSS Version 25. Participants were categorised into three groups: non-shielders (controls), participants primarily shielding themselves, and participants shielding others. The control group included those who had shielded in the past, as analyses showed that on key study variables, these individuals were most similar to non-shielders. Pearson’s correlations were used to examine associations between the main study variables, and demographics were explored using Pearson’s correlations and independent samples *t*-tests.

To examine the COVID-19 behaviour scale in more detail, participants who reported performing behaviours on average less than government guidance (i.e., mean scores of 2.5 and below), were compared with participants who reported performing behaviours more than government guidance (i.e., mean scores of 3.5 and above) using a Mann–Whitney U test. A non-parametric test was used as the data was not normally distributed, and there were some outliers. One sample *t*-tests were used to compare levels of generalised anxiety and health anxiety in both non-shielders and those shielding themselves with UK data from the first wave of the pandemic [9] (*N* = 842).

The differences in anxiety levels, contamination fears, and COVID-19-related behaviours between the three groups (shielding others, shielding self, and control) and two types of vaccination status were assessed using a two-way multivariate analysis of variance. All assumptions were met apart from the assumption of homogeneity of variances, which had been violated for three of the four dependent variables. This was corrected using square root transformations to the generalised anxiety and health anxiety variables and an inverse transformation on the contamination data (as this variable was strongly positively skewed).

Two multiple hierarchical linear regressions were used to explore whether contamination fears and specific COVID-19 behaviours predicted anxiety levels, with gender, age, friend/family member having had COVID-19, previous mental health condition, and vulnerable group status controlled for in order to isolate the effects of contamination fears and COVID-19 health behaviours. There were no significant differences between anxiety levels in those who were vaccinated and unvaccinated, GAD-7 *t*(720) = −0.69, *p* = 0.491; HAI *t*(719) = 0.61, *p* = 0.540, so this demographic was not controlled for in the analysis. Dummy variables were created for all categorical variables. All assumptions (independence and normal distribution of residuals, homoscedasticity, and no multicollinearity between independent variables) were met.

Finally, the relationship between vicarious health anxiety (i.e., worry about others) and own levels of generalised anxiety and health anxiety in the ‘shielding others’ group was assessed using Pearson’s correlations. A regression was not used because preliminary analysis of study demographic data indicated that there were no demographic variables significantly related to vicarious health anxiety levels, so no variables needed to be controlled for in the analysis. Inspecting boxplots for the four dependent variables indicated two outliers for GAD-7; however, because these outliers were not extreme, they were kept in the analysis. The difference scores for the dependent variables were normally distributed, as assessed by visual inspection of a Normal Q–Q plot.

## 3. Results

### 3.1. Relationship between Key Study Variables and Demographic Variables

There was a significant, positive relationship between all cross-group variables, indicating that, generally, participants who were health anxious also had high levels of generalised anxiety, fear of contamination (FoC) and excessive use of COVID-19-related behaviours. However, in the shielding others group, vicarious health anxiety was only significantly correlated with levels of generalised anxiety and FoC (see Table 3).

Analyses of the demographic variables indicated that there was a positive, significant relationship between participants’ ages and their health anxiety scores (i.e., older participants being more health anxious; *r*(708) = 0.08, *p* = 0.035). However, the opposite was true for generalised anxiety, with those younger having significantly higher generalised anxiety scores (*r*(709) = −0.10, *p* = 0.010). Females had significantly higher scores than males on both anxiety measures (HAI *t*(148.73) = −4.12, *p* < 0.001; GAD *t*(153.30) = −3.79, *p* < 0.001), and there was no significant difference between health anxiety and generalised anxiety scores across different ethnic groups (HAI *p* = 0.146; GAD *p* = 0.232). When comparing those with and without pre-existing mental health conditions, those who reported difficulties displayed significantly higher levels of anxiety than those who did not (HAI *t*(427.85) = 7.99, *p* < 0.001; GAD *t*(446.38) = 8.71, *p* < 0.001).

For COVID-19 diagnoses, there was no significant difference in anxiety scores when comparing those who had and had not been diagnosed with COVID-19 (HAI *p* = 0.748; GAD *p* = 0.302). However, individuals who had a close friend or family member who had been diagnosed with COVID-19 were significantly more anxious than the rest of the sample (HAI *t*(704) = 2.18, *p* = 0.029; GAD *t*(705) = 2.70, *p* = 0.007).

Finally, for those shielding themselves, there was a positive significant relationship between the number of months they had shielded for and participants’ levels of health anxiety (*r*(377) = 0.11, *p* = 0.030). The relationship between generalised anxiety and the number of months shielding was non-significant (*p* = 0.129).

The relationship between vicarious health anxiety and the demographic variables in the shielding others group was also explored. Findings showed no significant relationships or differences (age *p* = 0.969; gender *p* = 0.961; ethnicity *p* = 0.446; previous mental health diagnosis *p* = 0.970; COVID-19 diagnosis self *p* = 0.974; COVID diagnosis other *p* = 0.202).

### 3.2. Prevalence of Health Anxiety and Generalised Anxiety in Shielders, Their Family Members and the General Population

Table 4 summarises the mean questionnaire scores for the key study variables across the three groups. A two-way multivariate analysis of variance showed that there was no significant interaction effect between group and vaccination status on the dependent variables (*F*(8, 1388) = 1.01, *p* = 0.429, Wilks’ Λ = 0.989; partial η^2^ = 0.006). The main effect of vaccination status was not significant (*F*(4, 694) = 1.07, *p* = 0.371, Wilks’ Λ = 0.994; partial η^2^ = 0.006), but the effect of group was significant (*F*(8, 1388) = 20.23, *p* < 0.001, Wilks’ Λ = 0.802; partial η^2^ = 0.104). Follow-up univariate analyses of variance using a Bonferroni-adjusted *p* value of 0.001 found that there were statistically significant differences in generalised anxiety (*F*(2, 697) = 18.72, *p* < 0.001, partial η^2^ = 0.05), health anxiety (*F*(2, 697) = 47.40, *p* < 0.001, partial η^2^ = 0.12), FoC, (*F*(2, 697) = 8.58, *p* < 0.001, partial η^2^ = 0.02), and use of COVID-19-related behaviours (*F*(2, 697) = 45.55, *p* < 0.001, partial η^2^ = 0.12). Using Cohen’s [40] criteria for partial η^2^, the effect size was small for FoC and generalised anxiety and medium for health anxiety and use of COVID-19-related behaviours.

Tukey post-hoc tests showed that the ‘shielding self’ group had significantly higher generalised anxiety and FoC than the non-shielders (*p* < 0.001) but that the ‘shielding others’ group was not significantly different from either the non-shielders (GAD-7 *p* = 0.238; PI-10 *p* = 0.074) or ‘shielding self’ group (GAD-7 *p* = 0.065; PI-10 *p* = 0.951). Health anxiety scores were significantly higher in the ‘shielding self’ group in comparison to the non-shielders (*p* < 0.001) and ‘shielding others’ (*p* < 0.001), and there was no difference between the non-shielders and ‘shielding others’ (*p* = 0.910). Finally, the use of COVID-19-related behaviours was significantly lower in non-shielders in comparison to the two shielding groups (*p* < 0.001), and there was no difference in safety behaviour use between the two shielding groups (*p* = 0.753).

In the ‘shielding self’ group, 37.7% of individuals displayed clinical levels of generalised anxiety, in comparison to 23.2% of those shielding others and 15.9% of non-shielders. For health anxiety, 40% of individuals shielding themselves met the clinical cut-off, in comparison to 11.6% of those shielding others and 12.5% of non-shielders. Although health anxiety regarding own health in the ‘shielding others’ group was generally low, 50.7% of this group met the clinical cut-off for levels of vicarious health anxiety if a standard HAI cut-off is applied (>18).

In comparison to our study published in the first wave of COVID-19 [9], non-shielders in the current study had significantly lower levels of generalised anxiety, *t*(263) = −2.38, *p* = 0.018, and there was no significant difference in levels of health anxiety *t*(263) = 1.96, *p* = 0.051. However, levels of anxiety had increased for shielders since the first wave, with significantly higher levels of generalised anxiety, *t*(389) = 2.16, *p* = 0.032, and health anxiety, *t*(388) = 7.95, *p* < 0.001 [9].

### 3.3. Fear of Contamination and COVID-19-Related Behaviours

The median scores for the COVID-19 behaviour scale items are shown in Table 5. Generally, participants performed most behaviours in line with government guidance, apart from avoiding others outside the household, which was, on average, more than government guidance.

Two Mann–Whitney U tests were run to determine whether there were significant differences in anxiety scores between participants who reported performing COVID-19-related behaviours on average below government advice and those who reported performing behaviours above guidance. Results showed that for both health anxiety and generalised anxiety, median scores for the above guidance group (HAI = 16, GAD-7 = 8) were significantly higher than median scores for the below average group (HAI = 11, GAD-7 = 4; HAI *U* = 5163.00, *z* = −4.99, *p* < 0.001; GAD-7 *U* = 4912.50, *z* = −5.40, *p* < 0.001).

Two hierarchical linear regressions were run to determine whether COVID-19-related behaviours and FoC were significant predictors of anxiety over and above control variables (age, gender, previous mental health diagnosis, shielding group status, and family/friend having COVID-19).

### 3.4. Generalised Anxiety

See Table 6 for a full summary of each hierarchical regression model predicting generalised anxiety. Alone, control variables accounted for 16.7% of the variance.

The full model (model two) included predictors of interest (COVID-19-related behaviours and FoC) in addition to control variables, and was statistically significant *R*^2^ = 0.24, *F*(2, 678) = 18.96, *p* < 0.001. The addition of the variables of interest accounted for an additional 7.1% of the variance, however, whilst FoC contributed significantly to the model (*B* = 0.26, *p* < 0.001), COVID-19-related behaviours did not (*B* = 0.04, *p* = 0.356).

For the GAD-7 regression, coefficients for the dummy variables indicated that those who indicated a pre-existing mental health difficulty scored significantly higher than those who indicated they did not (*B* = 2.499, *SE* = 2.783, *p* < 0.000), and those who were shielding themselves and others scored significantly higher than those who were not shielding at all (*B* = 2.239, *SE* = 0.466, *p* < 0.000; *B* = 1.396, *SE* = 0.690, *p* = 0.004, respectively).

### 3.5. Health Anxiety

See Table 7 for a full summary of each regression model predicting health anxiety. Alone, control variables accounted for 18.9% of the variance. The full model (model two) included key predictors of interest (COVID-19-related behaviours and FoC) and was statistically significant *R*^2^ = 0.28, *F*(2, 677) = 23.25, *p* < 0.000. The addition of the variables of interest accounted for an additional 9% of the variance, however, whilst FoC contributed significantly to the model (*B* = 0.29, *p* < 0.000), COVID-19-related behaviours did not (*B* = 0.05, *p* = 0.251). The coefficients for the dummy variables indicated that those who indicated a pre-existing mental health difficulty or preferred not to say scored significantly higher than those who indicated they did not (*B* = 2.724, *SE* = 0.519, *p* < 0.000; *B* = 4.023, *SE =* 1.276, *p* < 0.002, respectively) and those who were shielding themselves scored significantly higher than those who were not shielding at all (*B* = 3.276, *SE* = 0.595, *p* < 0.000). This is consistent with tests of association and tests of difference.

## 4. Discussion

This study examined the prevalence and nature of psychological distress in those shielding during the second wave of the COVID-19 pandemic, building on previous research which highlighted the outstanding psychological need in ‘vulnerable’ groups in comparison to the general population [9]. Our findings make a novel contribution to the pandemic literature in examining the mental health of the clinically vulnerable on a larger scale and in more psychological depth, but also in assessing distress in those who are shielding others.

Consistent with data from previous waves, results reflected that psychological distress was elevated and ongoing in vulnerable groups [9,13,41,42,43,44,45]. Unsurprisingly, those shielding themselves were more distressed and fearful of contamination and reported higher levels of recommended health behaviours in comparison to those shielding others or not shielding at all, which is consistent with previous literature on risk factors for poor mental health during the pandemic [19]. In relation to distress and nature of distress, subgroups also emerged: those in older populations demonstrated higher rates of health anxiety than those younger, while the latter reported higher rates of generalised anxiety; females and those with previous mental health problems showed higher levels of distress, also consistent with previous findings [9,46,47,48,49]. Common multimorbidity in older populations and the susceptibility of those with comorbidity to experiencing health anxiety [50] are likely to contribute to this distress, particularly as COVID-19 mortality statistics continue to reflect high rates of mortality in those who are older [51], have underlying medical conditions [52], and those with medical complaints are more health anxious than the general population [53,54]. Previous explanations of a higher prevalence of anxiety in females attribute differences to multiple factors, including increased likelihood of difficult or traumatic life experiences [55,56], differences in coping styles between males and females [57,58,59], and also the impact of a higher proportion of self-selecting females participating in research [60,61]. With regards to younger females, in particular, recent research suggests that societal pressures may play a significant role [62,63], with the early 20s also commonly representing a period of help-seeking behaviour, which may reflect a peak prior to accessing intervention and support [64].

Exposure, rather than the diagnosis of COVID-19, was associated with higher levels of generalised anxiety. This may be explained by distress commonly associated with uncertain circumstances [9,10], which is particularly salient to this group due to the high likelihood of negative consequences of COVID-19 in the vulnerable, both in terms of mortality and particularly as we see those hospitalised with COVID-19 bear significant impact on their mental [65] and physical health [66].

In comparison to our previous work in the first wave, generalised anxiety of the broader population was lower, perhaps indicating some acclimatisation/habituation to the health threat or moderation of fear as (effective) treatments emerged. On the contrary, levels of generalised and health anxiety in the shielding population increased, with the length of time shielding correlated positively with health anxiety, despite increased knowledge and treatment over the course of time. Existing models of anxiety reflect that severity of anxiety is related to the degree of perceived threat, the severity of the implications, coping, and resources available to respond to the threat [67]; it is possible that anxiety in the general public may have reduced for these reasons. However, for those who are shielding, anxiety may sustain or indeed increase over time as the severity of the threat continues, and coping strategies continue to be at least partly restricted.

Those shielding themselves experienced higher levels of fear of contamination and were also over-engaging with COVID-19-related health behaviours, more so than the other groups. On these same measures, those shielding others were elevated but more similar to those who were not shielding. This is consistent with the cognitive behavioural model of health-related anxiety [67], which posits that threat-based cognition, such as fear of contamination, informs behavioural responses which are primarily designed to manage the ‘threat’; however, the model posits that ‘excessive’ or unhelpful behaviours are likely to reinforce threat beliefs and increase anxiety [68]. This is mirrored in the COVID-19 uncertainty model, which also highlights the key relationship between threat perception and over-engagement of behaviours directed at reducing uncertainty [10]. The specificity of the contribution of these two factors to distress in this study replicates previous pandemic research, reporting fear of contamination as a key predictor accounting for distress in other pandemics, e.g., Ebola [69], avian flu [70], and swine flu [22]. Despite using various methods of measurement, behavioural responses were not identified as a significant independent contributor to distress in any of these studies, yet significant correlations in expected directions in this and the aforementioned studies. This is difficult to unpick: studies have been well-powered, and conceptually, behaviour is implicated in our understanding of the persistence of anxiety. It is possible that what was measured in this study was behavioural compliance to pre-specified guidance, rather than idiosyncratic ‘safety-seeking’ behaviours, i.e., the measure may not have adequately captured *excessive* behaviours which are known to perpetuate distress in the anxiety model; or as observed in Blakey et al. [69]. An alternative view by Blakey et al. [69] suggests that anxiety in a pandemic may operate differently, taking the form of transient anxiety based on a realistic appraisal of threat to life, or simply a more complex relationship which could not be unpacked with our data or our purpose-developed measure.

On average, most people indicated they followed the precise government guidance required, but there was variation. Work by Drury et al. [71] indicated that one of the key factors associated with adherence to public health behaviour during the COVID-19 pandemic was empathy with vulnerable groups and the investment of a shared experience; a very salient notion to those shielding themselves/others and consistent with the idea that in a context of uncertainty, ‘rules’ offer some welcome certainty.

One of the key study findings was the incidence of health anxiety for others: over half of participants shielding others were experiencing high levels of vicarious health anxiety. Yet, they did not exhibit health anxiety for their own health, presumably as their perceived comparative risk was assessed to be substantially lower. Excellent internal consistency and psychometric evaluation reflect that not only is this vicarious phenomenon present but that many during the pandemic have been affected. Few studies exist in this area to date, with all published studies relating to parental anxiety over a child’s health [72,73,74,75]. This goes beyond the work examining the transmissibility of anxiety [76,77,78] to identifying a phenomenon where another person’s health is the primary concern, causing clinical levels of distress. This offers more specificity than previously seen in cognate fields such as caregiver literature. Previous research has highlighted generalised anxiety and health anxiety are related but distinct constructs [79,80], with tests of convergent validity in this study further confirming vicarious health anxiety as related but distinct from both health anxiety (*p* = 0.423) and generalised anxiety (*p* = 0.486). These findings provide a robust rationale for exploring this emerging concept of health on behalf of another, both in those who shield but also more broadly in those who care for others.

### 4.1. Clinical Implications

Adaptation of existing models of CBT may be beneficial to those who are shielding themselves or others, specifically focusing on realistic contamination-related fears and excessive or over-engaged behavioural responses. While cognition and behavioural components form central aspects of any CBT intervention, the specificity of these dimensions will increase the likelihood of improved outcomes [81,82,83]. Existing interventions targeting contamination fears might usually focus on exposure response prevention and elimination of safety-seeking behaviours; however, this must be carefully managed during a pandemic or in the context of realistic concerns. What may prove more useful are interventions to enable recalibration of contamination-based threat appraisals, targeting tolerance of the uncertainty produced by a novel virus and changing government guidelines, and how to respond to this behaviourally for both those who are clinically vulnerable and those seeking to protect others. This may also be helpful in preparation for future pandemics and infectious disease outbreaks and settings.

### 4.2. Limitations

As reported elsewhere, conducting research rapidly in response to a significant event invites vulnerability in the robustness of the design and delivery of research [84]; while there were no suitable existing measures that reliably measured behaviours in the way that was required, this study did not produce a measure with high internal consistency. Using relatively small numbers of items can induce vulnerability to a type 2 error [85], and bias towards more socially desirable responses may also have further weakened the reliability of the measure [86]. Rigorous procedures are in place to support the development of psychological measurement [87], and the demands of time sensitivity and robustness need to be weighed carefully. There are inherent limitations to using a self-report cross-sectional design, particularly in the context of rapid changes and novel contexts: self-report rates of mental health issues are known to be vulnerable to artificial inflation [88], particularly in a self-selecting online sample. Yet, findings replicate existing research and offer new findings. Notwithstanding these limitations, the findings bear clinical utility and have established areas worthy of further investigation: vicarious health anxiety and the mechanisms of this potentially reciprocal relationship.

### 4.3. Directions for Future Research

Future research should continue to monitor and increase understanding of the long-term impact and psychological needs of those shielding during the COVID-19 pandemic; we have repeatedly established that this neglected group bears a substantial burden due to their clinical circumstances. Further exploration of vicarious health anxiety and what may underpin the development, maintenance, and treatment of health anxiety is also warranted, offering a new avenue for understanding the distress experienced on behalf of another. Finally, it is evident that we are yet to fully understand behaviour in the context of a pandemic; an in-depth analysis of the motivations and cognitions underpinning behaviour is necessary to more adequately and reliably measure the impact of behavioural responses, particularly if this is a key factor within a dyad of mutually maintained health-related anxiety.

## 5. Conclusions

Those shielding during the pandemic have experienced sustained levels of distress. Special consideration must be given to those who are experiencing distress associated with the health of a vulnerable loved one, in addition to those more directly at risk. Psychological interventions should account for the realistic fear of contamination and the impact of government-recommended health behaviours, as these factors are associated with distress in vulnerable groups and may extend beyond the pandemic. Further research exploring vicarious health anxiety is warranted.

## Figures and Tables

**Table 1 ijerph-19-07333-t001:** Demographics for all included participants.

	Total		Shielding Self		Shielding Others		Non-Shielders	
Demographic	*n*	%	*n*	% *of Group*	*n*	% *of Group*	*n*	% *of Group*
*Gender*								
Male	109	15.1	42	10.8	26	37.7	41	15.5
Female	609	84.2	347	89.0	42	60.9	220	83.3
Other/prefer not to say	5	0.7	1	0.3	1	1.4	3	0.4
*Ethnicity*								
White	676	93.5	369	94.6	66	95.7	241	91.3
Mixed/multiple ethnic groups	15	2.1	10	2.6	1	1.4	4	1.5
Asian/Asian British	25	3.5	6	1.5	2	2.9	17	6.4
Black/African/Caribbean/Black British	2	0.3	2	0.5	0	0	0	0
Other ethnic group	5	0.7	3	0.8	0	0	2	0.8
*Employment*								
Working (full-time)	292	40.4	139	35.6	33	47.8	120	45.5
Working (part-time)	132	18.3	88	22.6	8	11.6	36	13.6
Not working (furloughed)	30	4.1	21	5.4	2	2.9	7	2.7
Not working (looking for work)	10	1.4	6	1.5	1	1.4	3	1.1
Not working (disabled)	51	7.1	42	10.8	6	8.7	14	5.3
Not working (retired)	62	8.6	48	12.3	2	2.9	1	0.4
Not working (other)	45	6.2	28	7.2	8	11.6	9	3.4
Prefer not to say	5	0.7	3	0.8	2	2.9	0	0
Student	92	12.7	13	3.3	6	8.7	73	27.7
Missing	4	0.6	2	0.5	1	1.4	1	0.4

**Table 2 ijerph-19-07333-t002:** Physical health conditions of those shielding themselves.

	Total	Shielding Self	Shielding Others	Non-Shielders
Categories	*n*	*n*	*n*	*n*
Aged 70 or older	33	19	9	5
Chronic respiratory diseases	238	193	20	25
Chronic heart disease	22	15	6	1
Chronic kidney disease	25	22	3	0
Chronic liver disease	10	10	0	0
Chronic neurological conditions	22	16	5	1
Diabetes	49	40	4	5
Problems with spleen	19	18	1	0
Weakened immune system	199	167	16	16
Seriously overweight (body mass index ≥ 40)	34	30	3	1
Pregnant	9	6	0	3
Other	78	56	12	10

**Table 3 ijerph-19-07333-t003:** Pearson’s correlations between main study variables.

Variables	1.	2.	3.	4.	5.
**1. HAI total**	-				
**2. GAD-7 total**	0.48 **	-			
**3. PI-10 total**	0.39 **	0.36 **	-		
**4. COVID-related behaviours total**	0.26 *	0.21 **	0.44 **	-	
**5. vHAI total (shielding others only)**	0.24	0.44 **	0.33 **	0.23	-

* Correlation is significant at the 0.05 level (2-tailed). ** Correlation is significant at the 0.01 level (2-tailed).

**Table 4 ijerph-19-07333-t004:** Summary statistics for groups.

Questionnaire	Normative Values	Shielding Self *N* = 390	Shielding Others *N* = 69	Non-Shielders *N* = 264
	*M (SD)*	*M (SD)*	*M (SD)*	*M (SD)*
GAD-7	6.42 (5.09)	8.18 (5.62)	6.77 (5.38)	5.41 (4.68)
HAI	9.19 (4.86)	16.34 (6.96)	10.70 (5.47)	11.31 (6.45)
PI-10	24.0 (23.6)	23.97 (10.58)	22.59 (9.55)	19.88 (8.12)
CRB	-	23.09 (3.66)	23.43 (3.60)	20.01 (3.46)
vHAI	-	-	18.52 (7.71)	-

Note. PI-10 = Padua Inventory 10-item contamination subscale; GAD-7 = 7-item General Anxiety Disorder; HAI = Short Health Anxiety Inventory; vHAI = vicarious Health Anxiety Inventory; CRB = COVID-19-Related Behaviours Scale.

**Table 5 ijerph-19-07333-t005:** Medians of COVID-19 behaviour scale items.

Items	Median	IQR
Washing hands and using hand sanitizer	3	1
Wearing a face covering	3	2
Avoiding other outside of your household	4	2
Getting COVID-19 tests	3	0
Stocking up on essentials	3	1
Checking internet for information on COVID-19	3	2
Attending clinical appointments	3	1

**Table 6 ijerph-19-07333-t006:** Summary of hierarchical multiple regression for predicting generalised anxiety.

Variable	Model 1			Model 2		
	*Unstandardised* *Co-Efficients*		*CE Beta*	*Unstandardised* *Co-Efficients*		*CE Beta*
	*Beta*	*SE Beta*		*Beta*	*SE Beta*	
Age	−0.06	0.01	−0.16	−0.05	0.01	−0.15
Shielding (other)	2.12	0.70	0.12	1.40	0.69	0.08
Shielding (self)	2.96	0.46	0.27	2.24	0.47	0.21
Gender (other)	2.33	2.91	0.03	2.57	2.78	0.03
Gender (prefer not to say)	−5.15	3.52	−0.05	−6.38	3.38	-0.06
Gender (male)	−1.21	0.55	−0.08	−0.78	0.53	-0.05
Pre-existing mental health condition (prefer not to say)	2.07	1.04	0.07	1.82	1.00	0.06
Pre-existing mental health condition (yes)	2.84	0.42	0.25	2.50	0.41	0.22
Know others diagnosed with COVID-19 (unsure)	0.58	1.32	0.02	0.83	1.27	0.02
Know others diagnosed with COVID-19 (yes)	0.61	0.40	0.06	0.61	0.38	0.06
Fear of contamination				0.15	0.02	0.26
COVID-19-related behaviour				0.05	0.06	0.04
Adjusted *R*^2^		0.17			0.24	
*F* for change in *R*^2^		14.88			32.47	

**Table 7 ijerph-19-07333-t007:** Summary of multiple regression for predicting health anxiety.

Variable	Model 1			Model 2		
	*Unstandardised* *Co-Efficients*		*CE Beta*	*Unstandardised* *Co-Efficients*		*CE Beta*
	*Beta*	*SE Beta*		*Beta*	*SE Beta*	
Age	−0.00	0.02	−0.01	0.00	0.02	0.01
Shielding (other)	−0.28	0.91	−0.01	−1.37	0.88	−0.06
Shielding (self)	4.34	0.60	0.30	3.28	0.60	0.23
Gender (other)	4.95	3.77	0.05	5.29	3.56	0.05
Gender (prefer not to say)	4.68	4.56	0.04	2.82	4.32	0.02
Gender (male)	−1.69	0.71	−0.09	−1.06	0.68	−0.05
Pre-existing mental health condition (prefer not to say)	4.39	1.35	0.11	4.02	1.28	0.10
Pre-existing mental health condition (yes)	3.21	0.55	0.21	2.72	0.52	0.18
Know others diagnosed with COVID-19 (unsure)	−1.60	1.71	−0.03	−1.23	1.62	−0.03
Know others diagnosed with COVID-19 (yes)	0.59	0.52	0.04	0.59	0.49	0.04
Fear of contamination (PI-10)				0.21	0.03	0.29
COVID-19-related behaviour				0.08	0.07	0.05
Adjusted *R*^2^		0.19			0.28	
*F* for change in *R*^2^		17.08			43.41	

## Data Availability

The data sets generated and analyzed during the current study are available on by contacting J.D. on jd494@bath.ac.uk.

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
