# Peer review of "The Mental Health Impact of the COVID-19 Pandemic Second Wave on Shielders and Their Family Members"

_ijerph, 2022, doi:10.3390/ijerph19127333_

Round 1

Reviewer 1 Report

Thank you for the opportunity to review this study entitled “The Mental Health Impact of the COVID-19 Pandemic Second Wave on Shielders and their Family Members.” (ijerph-1726068).

The study focused on the psychological effect of the COVID-19 pandemic, by investigating the association between fear of contamination (FoC) and use of government recommended behaviours and psychological distress.

In my opinion, the research topic is relevant, and the study is interesting. Parallelly, there are some issues that need to be addressed before the paper will be suitable for publication.

  • Abstract: the information about the sample should be deepened (Mean age and SD? Percentage of men and women?) to provide a clear picture of what will be presented in the paper.
  • Abstract: In this section, indices can be avoided (e.g., α, p).
  • Introduction: In my opinion, it would be good to refer to trend or longitudinal studies, if any. Since the authors frame this study considering the impact that COVID-19 has on a psychological level on people, I suggest some research to propose a comprehensive framework in the introduction, which should be supplemented with further literature search by the authors:

- Hyland et al., 2021; doi: 10.1016/j.psychres.2021.113905.

- Gori & Topino, 2021; doi: 10.3390/ijerph18115651

To find the suggested articles, the authors can use this source: https://www.doi.org/

  • Method: the questionnaire used in this study should be described in a section entitled “measures”, separately from “Participants and procedure”.
  • Complementary to the limitations, directions for future research should be indicated.

In general, I really enjoyed this paper, which seems to be well structured, interesting, and pleasant to read. In my opinion, after the authors make small changes, it will be ready to be published.

Author Response

Thank you for your comments, we feel the amendments have greatly improved the paper. We have attempted to make amendments that fall within the scope and remit of the paper, but also holding in mind the length of the paper.

We would like to draw your attention to the regression tables (table 6 and 7) which have additional rows, and further comment about the dummy variables. Following reviewer comments, we identified that data pertaining to the dummy variables were missing. This has made no material difference to the outcomes in either regression.

Reviewer 1

  1. Abstract: the information about the sample should be deepened (Mean age and SD? Percentage of men and women?) to provide a clear picture of what will be presented in the paper.

Thank you we have now included these suggestions, plus the final sample size.

  1. Abstract: In this section, indices can be avoided (e.g., α, p).

Thank you for your comment. We note that other abstracts in the journal retain indices, therefore on advice we have retained the p and α values.

  1. Introduction: In my opinion, it would be good to refer to trend or longitudinal studies, if any.Since the authors frame this study considering the impact that COVID-19 has on a psychological level on people, I sugges t some research to propose a comprehensive framework in the introduction, which should be supplemented with further literature search by the authors:

- Hyland et al., 2021; doi: 10.1016/j.psychres.2021.113905.

- Gori & Topino, 2021; doi: 10.3390/ijerph18115651

To find the suggested articles, the authors can use this source: https://www.doi.org/

Thank you. We have reviewed the literature, and as there is an extensive body of papers exploring the mental health impact of COVID-19 on the general population, we have chosen to update the introduction with a summary of findings from two systematic recent reviews and a meta-analysis, rather than outlining findings from individual research papers. The findings from these papers are consistent with our previous work and continue to highlight a gap, therefore supporting our rationale but with further up to date references. Thank you for your suggestion.

References:

Vindegaard, N; Benros, ME. COVID-19 pandemic and mental health consequences: Systematic review of the current evidence. Brain, behavior, and immunity. 202089, 531-542. doi:10.1016/j.bbi.2020.05.048

Robinson, E; Sutin, AR; Daly, M; Jones, A. A systematic review and meta-analysis of longitudinal cohort studies comparing mental health before versus during the COVID-19 pandemic in 2020. J Affect Disord2022296, 567-576. doi:10.1016/j.jad.2021.09.098

  1. Method: the questionnaire used in this study should be described in a section entitled “measures”, separately from “Participants and procedure”.

Thank you, this was an oversight on our part, a new section has now been inserted.

  1. Discussion: Complementary to the limitations, directions for future research should be indicated.

Thank you for pointing this out, we have now inserted a section on future research directions:

4.3 Directions for future research

            Future research should continue to monitor and increase understanding of the long -term impact and psychological needs of those shielding during the COVID-19 pandemic; we have repeatedly established this neglected group bear substantial burden due their clinical circumstances. Further exploration of vicarious health anxiety, what may underpin the development, maintenance and treatment of health anxiety is also warranted, offering a new avenue for understanding the distress experienced on behalf of another. Finally, it is evident that we are yet to fully understand behaviour in the context of a pandemic; an in-depth analysis of the motivations and cognitions underpinning behaviour is necessary to more adequately and reliably measure the impact of behavioural responses, particularly if this is a key factor within a dyad of mutually maintained health related anxiety.

Reviewer 2 Report

The article is well-established and indicates an important perspective of the pandemic that has to do with the psychological impact which is within the general scope of the Journal.

Some minor revisions are needed before publishing:

1. In the section of the abstract some concluding remarks of the research project should be added.

2. Ethical Approval (line 135-136) should be added after the conclusion section at the section material and methods are not so appropriate. Instead, characteristics of the research sample and demographics that are explained afterward (lines 153-160) should be more convenient.

Author Response

Thank you for your comments, we feel the amendments have greatly improved the paper. We have attempted to make amendments that fall within the scope and remit of the paper, but also holding in mind the length of the paper.

We would like to draw your attention to the regression tables (table 6 and 7) which have additional rows, and further comment about the dummy variables. Following reviewer comments, we identified that data pertaining to the dummy variables were missing. This has made no material difference to the outcomes in either regression.

Reviewer 2 comments

  1. In the section of the abstract some concluding remarks of the research project should be added.

Thank you, we have now inserted additional remarks and expanded the final section as follows:

Those shielding during the pandemic have experienced sustained levels of distress; special consideration must be given to those indirectly affected. Psychological interventions should account for realistic FoC, with future research continuing to monitor and better understand the clinical need of those shielding others post-pandemic.  

  1. Ethical Approval (line 135-136) should be added after the conclusion section at the section material and methods are not so appropriate. Instead, characteristics of the research sample and demographics that are explained afterward (lines 153-160) should be more convenient.

Thank you for your comment. We have used the Journal approved template of where to place the ethical approval, so have retained the current position in line with their requirements.

Reviewer 3 Report

1. The full article is fluent, novel in intention, rigorous and effective in experimental design and experimental treatment.

2. The study adopted the questionnaire survey method and distributed online to all British citizens. The sample was well representative. Future studies can increase sample diversity and comprehensively investigate the differences between Eastern and Western.

3. About whether the painful symptoms of family members pointed out in the article are related to the time of mutual contact between family members, mutual evaluation of each other and whether the emotional connection of the baseline state is strong? How did this survey control for or exclude these factors?

4. What are the connections and differences among health anxiety, anxiety, vicarious trauma, and vicarious health anxiety that appeared in the article?

5. Regarding the findings of this article, are there any other ways to reduce or prevent the emergence of adverse emotions in such subjects of this survey?

6.The internal consistency of data in this study is not very good.

7.In terms of model construction, the path of government-recommended behavior factors and health anxiety is not significant, and the model fitting index is not good, but it is highly correlated with generalized anxiety. In the discussion section, possible causes can be further discussed.

8.This study found that vicarious anxiety is widespread in the population, but cognitive behavioral therapy, which is usually used to intervene, needs to be used cautiously in the epidemic. 

9.The results showed that older individuals scored significantly higher on health anxiety inventory (HAI), but young people scored higher on generalized anxiety disorder (GAD).However, the author only explained why older individuals may have higher Hai scores (chronic disease distress, medical resources, etc.), but did not explain why young people have higher general anxiety. In addition, the results show that women score significantly higher than men in both Hai and GAD, but this paper does not give a corresponding discussion.

Author Response

Thank you for your comments, we feel the amendments have greatly improved the paper. We have attempted to make amendments that fall within the scope and remit of the paper, but also holding in mind the length of the paper.

We would like to draw your attention to the regression tables (table 6 and 7) which have additional rows, and further comment about the dummy variables. Following reviewer comments, we identified that data pertaining to the dummy variables were missing. This has made no material difference to the outcomes in either regression.

Reviewer comments

  1. About whether the painful symptoms of family members pointed out in the article are related to the time of mutual contact between family members, mutual evaluation of each other and whether the emotional connection of the baseline state is strong? How did this survey control for or exclude these factors?

If we understand your comment correctly, we believe you are raising the question as to the reciprocal nature of the anxiety and timing of exposure, plus the bond between the two people. As this was the first exploration of the concept of vicarious health anxiety, this aspect of the study was very much preliminary in nature, therefore the scope was fairly focused in this respect. However, these findings do provide a strong basis for further research to explore the underpinning mechanisms, as we now state in the future research directions section:

4.3 Directions for future research

Future research should continue to monitor and increase understanding of the long -term impact and psychological needs of those shielding during the COVID-19 pandemic; we have repeatedly established this neglected group bear substantial bur-den due their clinical circumstances. Further exploration of vicarious health anxiety, what may underpin the development, maintenance and treatment of health anxiety is also warranted, offering a new avenue for understanding the distress experienced on behalf of another. Finally, it is evident that we are yet to fully understand behaviour in the context of a pandemic; an in-depth analysis of the motivations and cognitions underpinning behaviour is necessary to more adequately and reliably measure the impact of behavioural responses, particularly if this is a key factor within a dyad of mutually maintained health related anxiety.

We also here refer to the potential of future work using dyads. We initially hoped to recruit dyads within this study for analysis of the relationship, however unfortunately we were unable to recruit sufficient data for meaningful analysis (less than 15 dyads with incomplete data). We do have ongoing work looking at vicarious health anxiety which does indeed take account of the nature of the relationship between the participant with an illness and the participant carer.

Please note: we do not mention ‘pain/painful’ symptoms in this paper so cannot comment on this aspect directly, please do let us know if we have misunderstood you.  

  1. What are the connections and differences among health anxiety, anxiety, vicarious trauma, and vicarious health anxiety that appeared in the article?

As outlined in the discussion, health anxiety and anxiety are both conceptually and statistically distinct but related constructs: (line 500-515)

One of the key study findings was the incidence of health anxiety for others: over half of participants shielding others were experiencing high levels of vicarious health anxiety. Yet, they did not exhibit health anxiety for their own health, presumably as their perceived comparative risk was assessed to be substantially lower. Excellent internal consistency and psychometric evaluation reflects that not only is this vicarious phenomenon present, but that many during the pandemic have been affected. Few studies exist in this area to date, with all published studies relating to parental anxiety over a child’s health [60-63]. This goes beyond the work examining transmissibility of anxiety [64-66] to identifying a phenomenon where another person’s health is the primary concern, causing clinical levels of distress. This offers more specificity than seen in cognate fields such as caregiver literature. Previous research has highlighted generalised and health anxiety are related but distinct constructs [67,68] with tests of convergent validity in this study further confirming vicarious health anxiety as a related but distinct from both health anxiety (p = .423) and generalized anxiety (p = .486). These findings provide a robust rationale for exploring this emerging concept of health on behalf of another, both in those who shield but also more broadly in those who care for others.

Simply put, we know that anxiety is rooted in self-referential beliefs, (whether this is health focussed or otherwise) and we are presenting a data driven hypothesis that vicarious health anxiety is anxiety experienced by another person (shielder, in this circumstance) but directly associated with the wellbeing of another (the person being shielded). Further research is planned to explore the relevance of self-referential beliefs in vicarious health anxiety, however our work needs further conceptual development before commenting in greater depth on these findings. We can only observe what we have measured.  

In terms of trauma (which was not measured in our paper, but on which I have previously published research) and vicarious trauma, the latter is also usually associated with empathic engagement with trauma survivors and the experience of distress associated with someone else’s distress.

Whether these two concepts are also similar yet distinct is unclear; there appears to be a qualitative difference in that anxiety is a common experience and data suggests presents similarly regardless of the focus of anxiety, however exposure to “actual or threatened death, serious injury, or sexual violence” (as per the DSM 5 criteria) is likely to be very different experience to feeling traumatised by empathic engagement with a trauma survivor (vicarious trauma) as it is currently explained in the literature.   

We thank the reviewer for highlighting this parallel, and the astute observation made. We have considered whether there is scope to allude to this parallel in this paper, however we concluded that it is beyond the scope of this paper as this is not the primary focus, and to do this justice, we would need to conceptually draw out the complexity of it, which would not be in the interests of the paper, would not be based on data, only speculation and would compromise the integrity of the paper. However, we do thank the reviewer, we will give these parallels further thought.

  1. Regarding the findings of this article, are there any other ways to reduce or prevent the emergence of adverse emotions in such subjects of this survey?

Thank you for your comments, you again raise interesting and salient points.  

Based on your other recommended changes, we have suggested that future research should also focus on the development, maintenance and treatment of health anxiety and based on your comment we have expanded the clinical implications section to ensure the inclusion of those with vicarious health anxiety when anticipating the potential psychological impact of a pandemic:

What may prove more useful are interventions to enable recalibration of contamination-based threat appraisals, targeting tolerance of the uncertainty produced by a novel virus and changing government guidelines, and how to respond to this behaviorally for both those who are clinically vulnerable and those seeking to protect others – this may be also helpful in preparation for future pandemics and infectious disease out-breaks and settings.

Having considered further expansion, we think this goes beyond the scope (and length requirements/focus) of this paper and we plan to explore this in future work; by establishing health anxiety as a present and relatively common presentation, this paves the way for future research exploring all suggested avenues of research. We thank you for raising these.

  1. The internal consistency of data in this study is not very good.

Thank you for your comment. Of the five measures, four measures demonstrated excellent internal consistency, with alpha levels between 0.91 and .94. We are satisfied with the performance of these measures in this sample as reliably measuring the constructs.

It is correct that our purpose produce measure of government recommended behaviours did not demonstrate high reliability, only achieving an alpha level of .61, which is disappointing and we do acknowledge in the limitations:

As reported elsewhere, conducting research rapidly in response to a significant event invites vulnerability in the robustness of the design and delivery of research [73]; while there were no suitable existing measures that reliably measured behaviours in the way that was required, this study did not produce a measure with high internal consistency. Using relatively small numbers of items can induce vulnerability to a type 2 error [74], and bias towards more socially desirable responses may also have further weakened the reliability of the measure [75]. Rigorous procedures are in place to support the development of psychological measurement [76], and the demands of time sensitivity and robustness need to be weighed carefully.  

We feel that given the potential reasons for the low reliability (stated above) and the findings being entirely consistent with behaviour not being predictive of outcome, it is unlikely to make a material difference to the findings, i.e. we are unlikely to have made a type 1 error.  We have however inserted a future research directions sections which includes prioritisation of better understanding and measurement of pandemic related behaviour:

4.3 Directions for future research

Future research should continue to monitor and increase understanding of the long -term impact and psychological needs of those shielding during the COVID-19 pandemic; we have repeatedly established this neglected group bear substantial bur-den due their clinical circumstances. Further exploration of vicarious health anxiety, what may underpin the development, maintenance and treatment of health anxiety is also warranted, offering a new avenue for understanding the distress experienced on behalf of another. Finally, it is evident that we are yet to fully understand behaviour in the context of a pandemic; an in-depth analysis of the motivations and cognitions underpinning behaviour is necessary to more adequately and reliably measure the impact of behavioural responses, particularly if this is a key factor within a dyad of mutually maintained health related anxiety.

We hope that by acknowledging the limitations, highlighting findings are consistent with the relevant broader literature and offering explanation which may also apply to the broader literature, that we transparently report findings in appropriate context while not over-stating the meaningfulness of these findings.

  1. In terms of model construction, the path of government-recommended behavior factors and health anxiety is not significant, and the model fitting index is not good, but it is highly correlated with generalized anxiety. In the discussion section, possible causes can be further discussed.

Thank you for your comment. We are a little unsure of your specific query and wonder if there is a word missing/typographical error, so we will attempt to speak to a broad interpretation of your point. We agree that the model fitting index (by which we take to mean the R squared) for government recommended behaviours (GRB) is non-significant and does not explain very much more in the health anxiety model, however it is also the same in the generalized anxiety model. These findings mirror other papers, as noted in the discussion lines 500 to 515. The correlations between GRB with both health anxiety and generalized anxiety are also similar in terms of strength of relationship and significance level. This again is consistent across the findings and in comparison with other papers.

Health anxiety and generalized anxiety are moderately correlated (.48), however this indicates convergent validity, similar but distinct constructs, which we discuss in line 532-537:

Previous research has highlighted generalised and health anxiety are related but dis-tinct constructs [67,68] with tests of convergent validity in this study further confirm-ing vicarious health anxiety as a related but distinct from both health anxiety (p = .423) and generalized anxiety (p = .486).

We have checked the VIF and tests of collinearity, and there is no evidence of collinearity, as noted in line 300.

Fear of Contamination and GRB are also correlated, however a value of .44 again reflects distinct but related constructs, also in absence of collinearity. This is a standard interpretation of a relationship of this strength, particular as these two factors would be predictably linked; if fear of contamination increases, GRB are also likely to increase, and vice-verse, as per the CBT model.

On line 555-563 we discuss the limitations of the rapid development of a measure, which may explain the poor contribution to the model (despite being consistent with existing findings)

We hope this further detail adequately responds to your queries – please do expand if we have not met the point.

  1. This study found that vicarious anxiety is widespread in the population, but cognitive behavioral therapy, which is usually used to intervene, needs to be used cautiously in the epidemic. 

Online and telephone facilitated CBT prior to the pandemic has been established as comparable in terms of efficacy (e.g. Spek V, Cuijpers P, Nyklı´cek I, Riper H, Keyzer J, Pop V. Internet-based cognitive behaviour therapy for symptoms of depression and anxiety: a meta-analysis. Psychol Med 2007; 37: 319–28; v Gilbody, S., Brabyn, S., Lovell, K., Kessler, D., Devlin, T., Smith, L., ... & Worthy, G. (2017). Telephone-supported computerised cognitive–behavioural therapy: REEACT-2 large-scale pragmatic randomised controlled trial. The British journal of psychiatry210(5), 362-367) and since the pandemic there has been a proliferation of papers and journal special editions on the adaptation of CBT for pandemic related difficulties (see: the Cognitive Behaviour Therapist / Volume 14 / 2021e8 accessed at https://www.cambridge.org/core/journals/the-cognitive-behaviour-therapist/special-issues/cbt-practitioner-guidance-for-during-and-following-the-covid-19-pandemic). Our conclusions are that as indicated in our paper, CBT should be adapted in order to accommodate unique aspects such as online therapy, adaptation of behavioural experiments and realistic fears of contamination (as we indicate below) however this only requires the usual CBT approach of an individualised formulation and adaptation of CBT derived from this:

Clinical implications (Line 517-530)

Adaptation of existing models of CBT may be beneficial to those who are shielding themselves or others, specifically focusing on realistic contamination-related fears and excessive or over-engaged behavioural responses. While cognition and behavioural components form central aspects of any CBT intervention, the specificity of these dimensions will increase the likelihood of improved outcomes [70-72]. Existing interventions targeting contamination fears might usually focus on exposure response prevention and elimination of safety seeking behaviours, however this must be carefully managed during a pandemic.

  1. The results showed that older individuals scored significantly higher on health anxiety inventory (HAI), but young people scored higher on generalized anxiety disorder (GAD).However, the author only explained why older individuals may have higher Hai scores (chronic disease distress, medical resources, etc.), but did not explain why young people have higher general anxiety. In addition, the results show that women score significantly higher than men in both Hai and GAD, but this paper does not give a corresponding discussion.

Thank you for highlighting this. We have now expanded on line 442 onwards to reflect the following:

Previous explanations of higher prevalence of anxiety in females attribute differences to multiple factors including increased likelihood of difficult or traumatic life experiences (Leach et al., 2008; McLean & Anderson; 2009) differences in coping styles between males and females (Kelly et al., 2008; Matud, 2004; Michl et al., 2013) and also the impact of higher proportion of self-selecting females participating in research (Jan & Mohajer, 2020; Porter & Whitcomb, 2005). With regards to younger females in particular, recent research suggests that societal pressures may play a significant role (Sherlock & Wagstaff, 2019; Wiklund et al., 2010) with early 20s also commonly represents a period of help-seeking behaviour, which may reflect a peak prior to accessing intervention and support (Mitchell et al., 2017). 

Round 2

Reviewer 3 Report

Some limitations are really difficult to change, but you have tried your best to answer the questions clearly.